# Ultrasound Imaging of the Articularis Genus Muscle: Implications for Ultrasound-Guided Suprapatellar Recess Injection

**DOI:** 10.3390/diagnostics14020183

**Published:** 2024-01-14

**Authors:** Wei-Ting Wu, Ke-Vin Chang, Ondřej Naňka, Kamal Mezian, Vincenzo Ricci, Bow Wang, Levent Özçakar

**Affiliations:** 1Department of Physical Medicine and Rehabilitation, National Taiwan University Hospital, Bei-Hu Branch, Taipei 10845, Taiwan; wwtaustin@yahoo.com.tw; 2Department of Physical Medicine and Rehabilitation, College of Medicine, National Taiwan University, Taipei 10048, Taiwan; 3Center for Regional Anesthesia and Pain Medicine, Wang-Fang Hospital, Taipei Medical University, Taipei 11600, Taiwan; 4Institute of Anatomy, First Faculty of Medicine, Charles University, 12800 Prague, Czech Republic; ondrej.nanka@lf1.cuni.cz; 5Department of Rehabilitation Medicine, First Faculty of Medicine and General University Hospital in Prague, Charles University, 12800 Prague, Czech Republic; kamal.mezian@gmail.com; 6Physical and Rehabilitation Medicine Unit, Luigi Sacco University Hospital, ASST Fatebenefratelli-Sacco, 20157 Milan, Italy; vincenzo.ricci58@gmail.com; 7Department of Medical Imaging, National Cheng Kung University Hospital, College of Medicine, National Cheng Kung University, No. 1 University Rd, Tainan 704302, Taiwan; wangbow1227@gmail.com; 8Department of Physical and Rehabilitation Medicine, Hacettepe University Medical School, Ankara 06100, Turkey; lozcakar@yahoo.com

**Keywords:** quadriceps, knee, ultrasonography, pain, hyaluronic acid

## Abstract

Elucidating its dynamic interaction within the knee joint, this exploration delves into the awareness regarding the articularis genus muscle for ultrasound-guided suprapatellar recess injections. While injections into the infrapatellar recess may proceed without ultrasound guidance, we highlight concerns regarding the potential cartilage injury. In contrast, especially with ultrasound guidance, suprapatellar recess injections significantly mitigate this risk, especially in the case of collapsed recess. Originating from the distal femur and vastus intermedius, the articularis genus muscle influences the tension of the suprapatellar recess during knee motion. Sonographically identifying this muscle involves visualizing the slender linear structure of the suprapatellar recess, with guidance on differentiation from the vastus intermedius. We provide a succinct approach to ultrasound-guided suprapatellar recess injections, emphasizing needle insertion techniques and strategies to prevent fluid accumulation. In conclusion, this study serves as a concise clinician’s guide, underscoring the significance of the articularis genus muscle’s sonoanatomy in ultrasound-guided suprapatellar recess injections. Ultimately, procedural precision and patient safety can be advanced in this aspect.

## 1. Introduction

The knee functions as a hinge joint, comprising three primary articulations formed by the femur, tibia, and patella. Various synovial recesses exist in the knee region, namely, the central synovial, suprapatellar, superior/inferior infrapatellar, femorotibial, retrocondylar, and sub-popliteal recesses [1]. The suprapatellar and superior/inferior infrapatellar recesses are frequently targeted as injection sites, allowing for the administration of substances like corticosteroids, hyaluronic acid, and platelet-rich plasma into the intra-articular space [2,3]. Injecting into the superior/inferior infrapatellar recess may not always require ultrasound guidance, and it can be performed by palpating the gap between the medial aspect of the patella and the medial distal femoral condyle. Therefore, when executed without ultrasound guidance, this method poses challenges regarding the potential cartilage injury of the distal femoral condyle.

Conversely, injecting through the suprapatellar recess significantly reduces the risk of distal femoral cartilage injury when performed under ultrasound guidance, particularly when the suprapatellar recess appears collapsed with minimal effusion [4]. Even with ultrasound guidance, the articularis genus muscle, a less-discussed structure, can introduce confusion regarding the precise location of the suprapatellar recess. Although there are magnetic resonance imaging (MRI) and cadaveric studies on the articularis genus muscle [5,6], there is currently a lack of publications that specifically investigate this muscle using ultrasound imaging. Therefore, this imaging essay aims to elaborate on the sonoanatomy of the articularis genus and its implications during ultrasound-guided suprapatellar recess injection.

## 2. Material and Methods

An experienced academic anatomist with over 25 years of expertise conducted the dissections. Two knees from formalin-fixed cadavers at the cadaveric laboratory of the First Faculty of Medicine, Charles University, Prague, were meticulously examined. Following a longitudinal skin incision at approximately the distal one-third of the knee, the skin was removed, and the suprapatellar recess and articularis genus were exposed through blunt dissection.

Concerning the histology section, the preparation involved the use of hematoxylin and eosin staining, in addition to Alcian blue, to unveil cartilage from the fetal period, particularly between the 50th and 80th day of development. These sections are part of the Doskočil’s collection housed at the Institute of Anatomy of the First Faculty of Medicine, Charles University, Prague. They were meticulously crafted in the 1960s and 1970s, adhering to the norms effective during that period.

MRI scans were acquired using the Signa Artist 1.5T whole-body imaging system manufactured by GE Healthcare Technologies, Inc. in Chicago, IL, USA. A 16-channel transmit/receive (T/R) knee coil designed for the GE 1.5T MRI system was employed for the imaging process. The scanning protocol covered a range extending 7.5 cm both proximally and distally from the level of the midpoint of the patella.

Ultrasound images were acquired utilizing the Aplio i600 platinum platform, an ultrasound system developed by Canon Medical System in Tokyo, Japan. The imaging process utilized linear transducers (PLT-1005BT, 58 mm wide, 3.8–10 MHz), with the scanning depth varying from 2 to 3 cm, and the focus predominantly set at 1.5 cm. A standardized frame rate of 30 frames per second was maintained throughout the imaging procedures. These tasks were performed by a highly skilled physician with over a decade of expertise in musculoskeletal ultrasound.

## 3. Anatomic Elaboration of Articularis Genus

The articularis genus emerges from the anterior bony cortex of the distal femur and the deep portion of the vastus intermedius, and attaches to the wall of the suprapatellar recess and the capsule of the knee joint [6]. Akin to the quadriceps muscle, it is innervated by the femoral nerve (L2–L4) and supplied from the lateral femoral circumflex artery. However, its precise function remains a topic of ongoing investigation. Some researchers suggest that the activation of the articularis genus elevates the wall tension of the suprapatellar recess, thereby protecting it from the potential impingement between the patella and the femur [5]. Grob et al. [5] have elucidated the dynamic alterations in the suprapatellar recess during knee motion, along with its correlation with the articularis genus muscle. When the knee is flexed, the articularis genus is elongated and remains inactive, exerting no traction force on the suprapatellar recess. Conversely, during knee extension, the articularis genus contracts, shortening and causing a pull on the suprapatellar recess wall and leading to its dilation (Figure 1).

Woodley et al. [6] have offered comprehensive insights into the articularis genus as a multi-layered muscle, aligning with the cadaveric images observed in our anatomical laboratory (Figure 2). Their study, based on cadaveric dissections, revealed an average of seven muscle bundles (ranging from four to ten) in the articularis genus. The mean cross-sectional area of the articularis genus was 1.5 cm^2^, with an average muscle bundle length of 5.9 cm. Notably, the articularis genus was found to consist of a combination of type I and type II muscle fibers.

Furthermore, in a cadaveric study conducted by Kimura et al. [7], the observation revealed varying amounts of fatty tissue interconnecting with the synovial membrane of the suprapatellar recess. Those fatty tissues acted as interpositions between each bundle of the articularis genus muscle, a finding consistent with our cadaveric histology specimen (Figure 3).

Additionally, a more substantial presence of fatty tissue was identified in the deeper portion of the articularis genus muscle [7]. This fatty infiltration is further highlighted in T1-weighted magnetic resonance imaging, where high-intensity fat tissues are discernible within the articularis genus muscle (Figure 4).

## 4. Sonoanatomy of Articularis Genus

To identify the articularis genus muscle sonographically, it is appropriate to begin by placing the transducer in the sagittal plane over the suprapatellar region with the knee extended (Figure 5A). The suprapatellar recess manifests as a slender linear structure, delineating the articularis genus from the overlying suprapatellar fat pad and vastus intermedius aponeurosis. Hyperechoic fat strips may be observed between the various bundles of the articularis genus muscle. Requesting the participant to fully extend the knee facilitates the visualization of the articularis genus muscle contraction, followed by the subsequent expansion of the suprapatellar recess (Figure 5B, Appendix A).

By adjusting the transducer proximally, the vastus intermedius muscle becomes visible cranial to the articularis genus muscle (Figure 6A). Using ultrasound imaging, the differentiation between the vastus intermedius and articularis genus muscles is achieved by examining their distal attachments. The former connects to the superficial aponeurosis, while the articularis genus muscle attaches to the suprapatellar recess, notably emphasized during the maximal knee extension (Figure 6B).

Once the vastus intermedius and articularis genus muscles are identified in their longitudinal axis, the transducer can be rotated to the short axis of the femur (Figure 7A). However, due to the non-horizontal interface between the vastus intermedius and the articularis genus, distinguishing the two muscles may pose a challenge. Progressing the transducer distally, the vastus intermedius muscle diminishes, allowing for an improved visualization of the suprapatellar recess situated between the articularis genus and the superficial aponeurosis of the vastus intermedius (Figure 7B).

## 5. Implication of Ultrasound-Guided Injection

Once the suprapatellar pouch is identified, the needle can be inserted using the in-plane technique either from lateral to medial or medial to lateral, depending on the convenience of the physician (Figure 8A). The needle used for injection was a 21-gauge needle with a length of 7 cm. This needle size was chosen to facilitate the administration of injectates with high viscosity, such as hyaluronic acid, and to target the central part of the suprapatellar bursa. In cases where there is no effusion to delineate the suprapatellar recess, the physician may gently mobilize the patella to ascertain the precise level of the suprapatellar pouch before administering the injection. Alternatively, the physician can apply slight pressure to the gap between the patella edge and the femur, observing the fat pad herniate into the suprapatellar recess. Ensuring that the needle is properly positioned within the recess guarantees an even distribution of the injectate within the slender suprapatellar recess, preventing fluid accumulation in a confined space (Figure 8B, Appendix A).

Additionally, a prior systematic review [8] underscored the superior accuracy of image-guided intra-articular knee injections at specific sites, including the superomedial patellar, medial midpatellar, superolateral patellar, and lateral suprapatellar bursae, compared to blind injections. However, there remains a knowledge gap regarding the long-term effectiveness of ultrasound-guided intra-articular knee injections compared to blind injections. A recent randomized controlled trial [9] demonstrated that ultrasound-guided infrapatellar injection led to an early and positive opaque transition to the knee joint in comparison to the suprapatellar approach, despite the clinical effects appearing similar between the two methods. Given that our report concentrates on a concise exploration of the sonoanatomy of the articularis genus and its relevance to guided injection, we have not conducted a formal study assessing the impact of detailing the sonoanatomy on the precision of intra-articular knee injections. This aspect could be investigated in future cadaveric and clinical studies.

## 6. Conclusions

In summary, this article underscores the significance of the frequently overlooked articularis genus muscle in the suprapatellar region. It emphasizes the importance of a physician’s awareness regarding the sonoanatomy of this muscle and its correlation with suprapatellar recess injections.

## Figures and Tables

**Figure 1 diagnostics-14-00183-f001:**
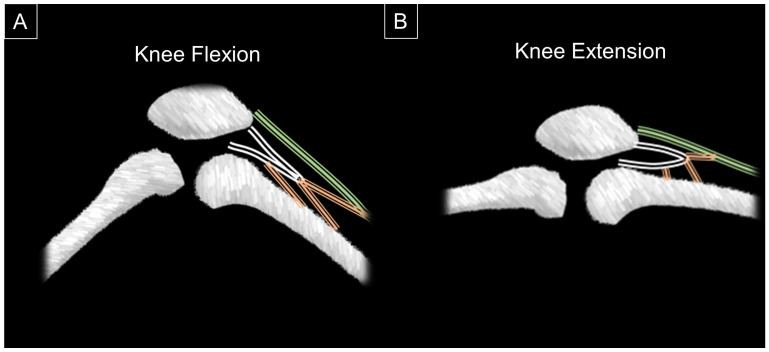
During knee flexion when the articularis genus (orange striated lines) and quadriceps (green striated line) remain inactive, reduced tension is exerted on the wall of the suprapatellar recess (white striated lines). This diminished tension causes the recess to collapse, resulting in a slender appearance (**A**). Conversely, with knee extension and the activation of the articularis genus, the tension on the wall of the suprapatellar recess intensifies. This increased tension leads to the dilation of the suprapatellar recess (**B**).

**Figure 2 diagnostics-14-00183-f002:**
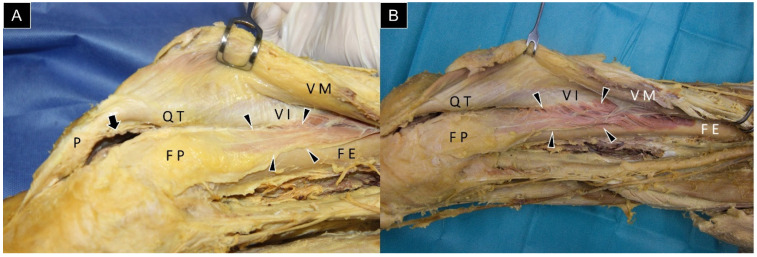
Cadaveric dissection images depict the articularis genus muscle (arrowheads) in different cases (**A**,**B**). Arrow, the existing portal of the suprapatellar recess. QT, quadriceps tendon; VI, vastus intermedius muscle; VM, vastus medialis muscle; FE, femur; FP, fat pad; and P, patella.

**Figure 3 diagnostics-14-00183-f003:**
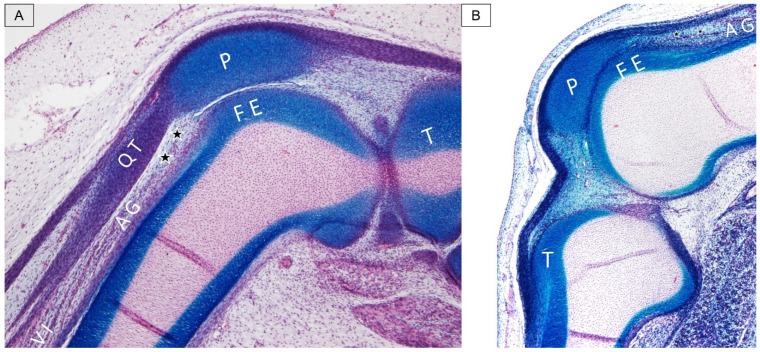
Cadaveric histology dissection images illustrate the articularis genus (AG) muscle in different cases (**A**,**B**). P, patella; FE, femur; T, tibia; QT, quadriceps tendon; VI, vastus intermedius; and stars, fat tissue.

**Figure 4 diagnostics-14-00183-f004:**
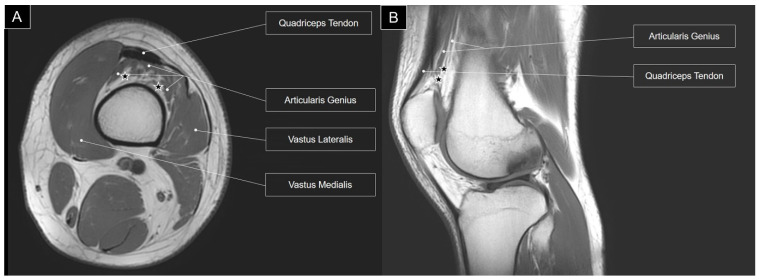
T1-weighted axial (**A**) and sagittal (**B**) views of magnetic resonance imaging for the articularis genus muscle. Stars, fat tissue.

**Figure 5 diagnostics-14-00183-f005:**
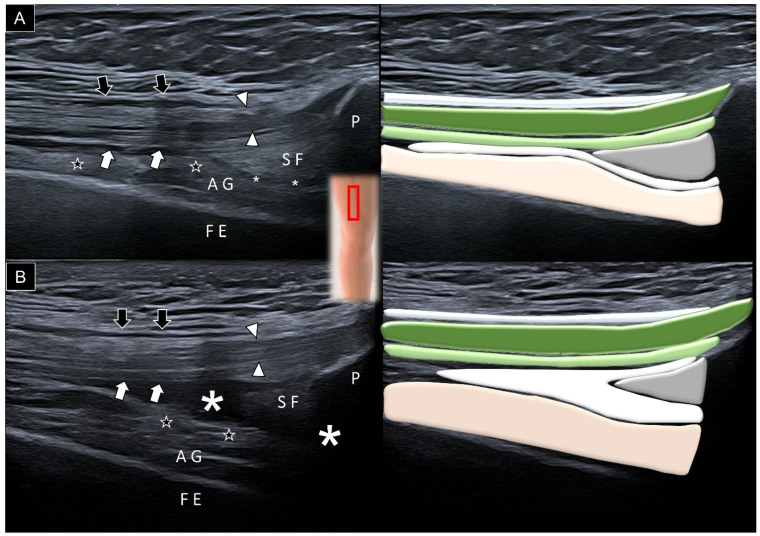
Sagittal ultrasound imaging and schematic drawing of the articularis genus (AG) muscle at the suprapatellar region during neutral knee extension (**A**) and maximal knee extension (**B**). Black arrows, aponeurosis from the rectus femoris; white arrowheads, aponeurosis from the vastus lateralis and medialis; white arrows, aponeurosis from the vastus intermedius; stars, fat tissue; asterisk, suprapatellar recess; FE, femur; P, patella; and SF, suprapatellar fat pad; red square, transducer’s location.

**Figure 6 diagnostics-14-00183-f006:**
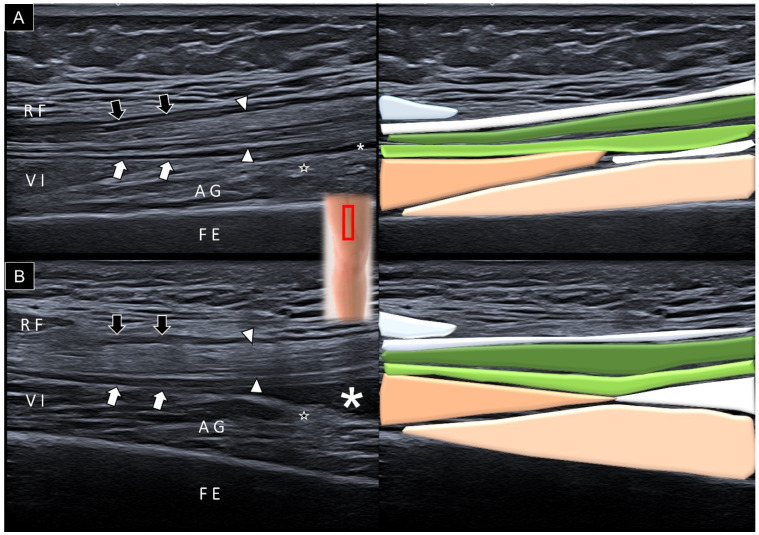
Sagittal ultrasound imaging and schematic drawing of the articularis genus (AG) muscle at the distal femoral region during neutral knee extension (**A**) and maximal knee extension (**B**), facilitating the visualization of vastus intermedius muscle (VI). Black arrows, aponeurosis from the rectus femoris; white arrowheads, aponeurosis from the vastus lateralis and medialis; white arrows, aponeurosis from the vastus intermedius; stars, fat tissue; asterisks, suprapatellar recess; FE, femur; RF, rectus femoris; red square, transducer’s location.

**Figure 7 diagnostics-14-00183-f007:**
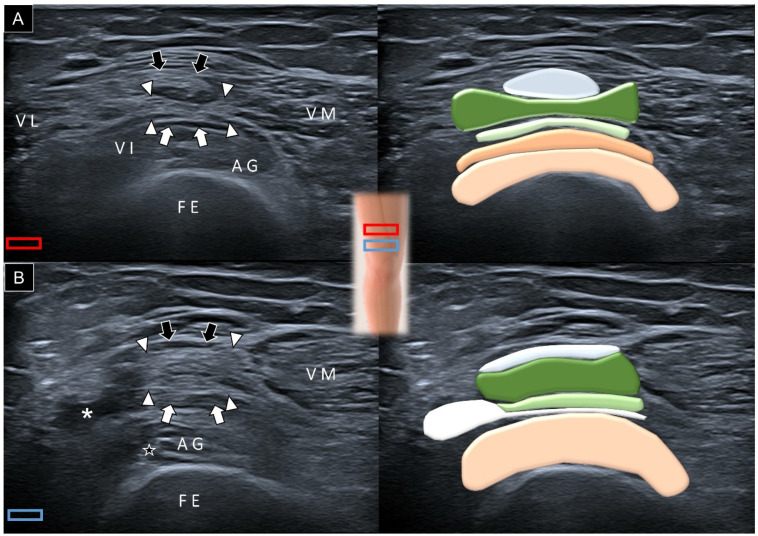
Axial ultrasound imaging and schematic drawing of the articularis genus (AG) muscle at the proximal femoral (**A**) and suprapatellar (**B**) regions. Black arrows, aponeurosis from the rectus femoris; white arrowheads, aponeurosis from the vastus lateralis and medialis; white arrows, aponeurosis from the vastus intermedius; star, fat tissue; asterisk, suprapatellar recess; FE, femur; P, patella; VI, vastus intermedius muscle; VM, vastus medialis muscle; VL, vastus lateralis muscle; red and blue squares, transducer’s location.

**Figure 8 diagnostics-14-00183-f008:**
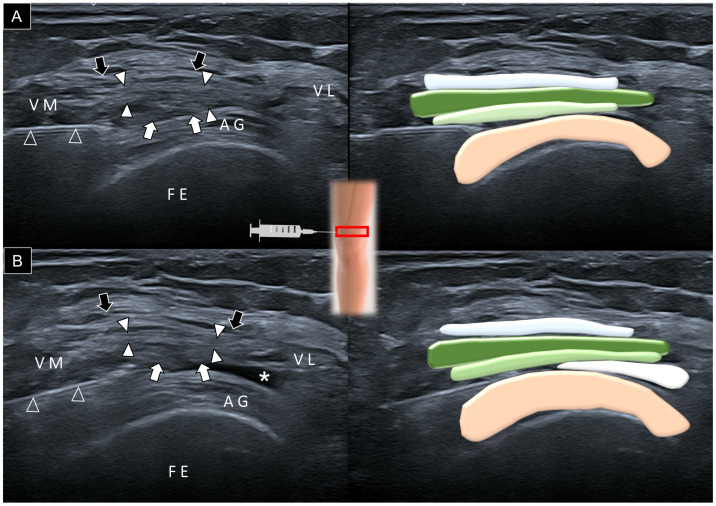
Ultrasound imaging and schematic drawing for the guided injection of the suprapatellar recess before (**A**) and after (**B**) introducing the injectate. Black arrows, aponeurosis from the rectus femoris; white arrowheads, aponeurosis from the vastus lateralis and medialis; white arrows, aponeurosis from the vastus intermedius; asterisk, suprapatellar recess; FE, femur; VM, vastus medialis muscle; VL, vastus lateralis muscle; AG, articularis genus muscle; red square, transducer’s location.

## Data Availability

Data are contained within the main text of the manuscript.

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
