# Peer review of "Ultrasound Imaging of the Articularis Genus Muscle: Implications for Ultrasound-Guided Suprapatellar Recess Injection"

_diagnostics, 2024, doi:10.3390/diagnostics14020183_

Round 1

Reviewer 1 Report

Comments and Suggestions for Authors

This manuscript reports ultrasound images of the articularis genus muscle. The manuscript belongs to the type of Interesting Images. I have not reviewed such a type before. The images are of high quality, in my point of view. And the manuscript is generally well-written. The manuscript does not contain any quantitative results. For a scientific paper, quantitative results are necessary. However, if the type of Interesting Images does not require quantitative results, then this concern can be neglected. In addition, please clarify the novelty of this work. Are these images not reported before?

Author Response

Reviewer 1

Comment:

This manuscript reports ultrasound images of the articularis genus muscle. The manuscript belongs to the type of Interesting Images. I have not reviewed such a type before. The images are of high quality, in my point of view. And the manuscript is generally well-written. The manuscript does not contain any quantitative results. For a scientific paper, quantitative results are necessary. However, if the type of Interesting Images does not require quantitative results, then this concern can be neglected. In addition, please clarify the novelty of this work. Are these images not reported before?

Response:

We acknowledge the reviewer's thoughtful comments. The article has been submitted to the "interesting imaging" session rather than the "review" or "original article" category. Consequently, there is no imperative to adhere to the conventional structural format with subtitles such as "introduction," "material and methods," "results," "discussion," and "conclusion."

For reference, we have provided examples of articles in the "interesting images" category from "Diagnostics" at the following links:

(1) https://www.mdpi.com/2075-4418/14/1/108

(2) https://www.mdpi.com/2075-4418/14/1/68

However, in response to the reviewer's concerns, we have incorporated the following subtitles to enhance the organizational structure of our article:

(1) Introduction

(2) Material and Methods

(3) Anatomic Elaboration of Articularis Genus 

(4) Sonoanatomy of Articularis Genus

(5) Implication of Ultrasound Guided Injection

(6) Conclusion

     Regarding the novelty of the present article, a sentence has been incorporated to emphasize: “Although there are magnetic resonance imaging and cadaveric studies on the articularis genus muscle, there is currently a lack of publications that specifically investigate this muscle using ultrasound imaging”.

Reviewer 2 Report

Comments and Suggestions for Authors

In this manuscript, the authors investigate the role of the articularis genus muscle in ultrasound-guided suprapatellar recess injections within the knee joints. Although injections into the infrapatellar recess can proceed without ultrasound guidance, the authors emphasize concerns about potential cartilage injury. Their findings provide a concise guide for clinicians, focusing on needle insertion techniques and strategies to prevent fluid accumulation, aiming to enhance procedural precision and patient safety. However, the manuscript lacks effective organization and has several issues that need addressing. It requires significant improvement for better clarity and flow. Here are some suggestions and comments.

(1)  In the methods section, it is important for the authors to outline the procedures for sample preparation, histology sectioning and staining, MRI imaging and ultrasound imaging.

(2)  Could the authors provide information on the ultrasound equipment utilized in the study? Additionally, what is the frequency of the probe used?

(3)  What is the needle size for intra-articular knee joints injections? Or how to choose the proper needle size for patients accordingly?

(4)  Could the authors include a discussion on the limitations of their study? And could the authors provide potential directions for future research in this area?

Author Response

Reviewer 2

General Comment:

In this manuscript, the authors investigate the role of the articularis genus muscle in ultrasound-guided suprapatellar recess injections within the knee joints. Although injections into the infrapatellar recess can proceed without ultrasound guidance, the authors emphasize concerns about potential cartilage injury. Their findings provide a concise guide for clinicians, focusing on needle insertion techniques and strategies to prevent fluid accumulation, aiming to enhance procedural precision and patient safety. However, the manuscript lacks effective organization and has several issues that need addressing. It requires significant improvement for better clarity and flow. Here are some suggestions and comments.

Response:

We acknowledge the reviewer's thoughtful comments. The article has been submitted to the "interesting imaging" session rather than the "review" or "original article" category. Consequently, there is no imperative to adhere to the conventional structural format with subtitles such as "introduction," "material and methods," "results," "discussion," and "conclusion."

For reference, we have provided examples of articles in the "interesting images" category from "Diagnostics" at the following links:

(1) https://www.mdpi.com/2075-4418/14/1/108

(2) https://www.mdpi.com/2075-4418/14/1/68

However, in response to the reviewer's concerns, we have incorporated the following subtitles to enhance the organizational structure of our article:

(1) Introduction

(2) Material and Methods

(3) Anatomic Elaboration of Articularis Genus 

(4) Sonoanatomy of Articularis Genus

(5) Implication of Ultrasound Guided Injection

(6) Conclusion

Specific Comment:

(1)  In the methods section, it is important for the authors to outline the procedures for sample preparation, histology sectioning and staining, MRI imaging and ultrasound imaging.

Response: We appreciate the invaluable feedback from the reviewer. Detailed descriptions of the procedures for sample preparation, histology sectioning and staining, MRI imaging, and ultrasound imaging have been provided as follows: “An experienced academic anatomist with over 25 years of expertise conducted the dissections. Two knees from formalin-fixed cadavers at the cadaveric laboratory of the First Faculty of Medicine, Charles University, Prague, underwent meticulous examination. Following a longitudinal skin incision at approximately the distal one-third of the knee, the skin was removed, and the suprapatellar recess and articularis genus were exposed through blunt dissection.

Concerning the histology section, the preparation involved the use of hematoxylin and eosin staining, in addition to Alcian blue, to unveil cartilage from the fetal period, particularly between the 50th and 80th day of development. These sections are part of the Doskočil´s collection housed at the Institute of Anatomy of the First Faculty of Medicine, Charles University, Prague. They were meticulously crafted in the 1960s and 1970s, adhering to the norms effective during that period.

MRI scans were acquired using the Signa Artist 1.5T whole-body imaging system manufactured by GE Healthcare Technologies, Inc. in Chicago, Illinois, USA. A 16-channel transmit/receive (T/R) knee coil designed for the GE 1.5T MRI system was employed for the imaging process. The scanning protocol covered a range extending 7.5cm both proximally and distally from the level of the midpoint of the patella.

Ultrasound images were acquired utilizing the Aplio i600 platinum platform, an ultrasound system developed by Canon Medical System in Tokyo, Japan. The imaging process utilized linear transducers (PLT-1005BT, 58 mm wide, 3.8-10 MHz), with the scanning depth varying from 2 to 3 cm, and the focus predominantly set at 1.5 cm. A standardized frame rate of 30 frames per second was maintained throughout the imaging procedures. These tasks were performed by a highly skilled physician with over a decade of expertise in musculoskeletal ultrasound.”

(2)  Could the authors provide information on the ultrasound equipment utilized in the study? Additionally, what is the frequency of the probe used?

Response:

We express our gratitude for the positive feedback provided by the reviewer. The ultrasound machine settings have been specified as follows: “Ultrasound images were acquired utilizing the Aplio i600 platinum platform, an ultrasound system developed by Canon Medical System in Tokyo, Japan. The imaging process utilized linear transducers (PLT-1005BT, 58 mm wide, 3.8-10 MHz), with the scanning depth varying from 2 to 3 cm, and the focus predominantly set at 1.5 cm. A standardized frame rate of 30 frames per second was maintained throughout the imaging procedures. These tasks were performed by a highly skilled physician with over a decade of expertise in musculoskeletal ultrasound”.

(3)  What is the needle size for intra-articular knee joints injections? Or how to choose the proper needle size for patients accordingly?

Response: In this article, the injection needle employed had a gauge of 21 and a length of 7 cm. This specific needle size was selected to facilitate the administration of injectates with high viscosity, such as hyaluronic acid, and to precisely target the central part of the suprapatellar bursa. The following sentence has been incorporated to emphasize this information: "The needle used for injection was a 21-gauge needle with a length of 7 cm. This needle size was chosen to facilitate the administration of injectates with high viscosity, such as hyaluronic acid, and to target the central part of the suprapatellar bursa."

(4)  Could the authors include a discussion on the limitations of their study? And could the authors provide potential directions for future research in this area?

Response: We are grateful for the reviewer's constructive feedback. It's important to note that this submission is not presented as a formal "study" but rather as part of the "interesting images" session. However, in response to the reviewer's concern, we have included a statement addressing the limitation: "Given that our report concentrates on a concise exploration of the sonoanatomy of the articularis genus and its relevance to guided injection, we have not conducted a formal study assessing the impact of detailing the sonoanatomy on the precision of intra-articular knee injections. This aspect could be investigated in future cadaveric and clinical studies.”

Reviewer 3 Report

Comments and Suggestions for Authors

There is an interesting description of the role of articularis genus muscle. However, the paper is more - or -less clearly written, there is no classic structure: introduction, material, etc. I would like to see short discussion why this finding is so important, ion comparison with the literature. More references are also necessary, ex. 

Maricar N, Parkes MJ, Callaghan MJ, Felson DT, O'Neill TW. Where and how to inject the knee--a systematic review. Semin Arthritis Rheum. 2013 Oct;43(2):195-203. doi: 10.1016/j.semarthrit.2013.04.010. Erratum in: Semin Arthritis Rheum. 2015 Apr;44(5):e18. Erratum in: Semin Arthritis Rheum. 2015 Apr;44(5):e18. PMID: 24157093; PMCID: PMC3820023.

Ertilav E, Sarı S, Ertilav D, Aydın ON. Comparison of radiological and clinical results of knee intra-articular injections with two ultrasonography-guided approach techniques: A randomized controlled study. Arch Rheumatol. 2022 Oct 21;38(2):230-237. doi: 10.46497/ArchRheumatol.2023.9382. PMID: 37680515; PMCID: PMC10481686.

Author Response

Reviewer 3:

Comment:

There is an interesting description of the role of articularis genus muscle. However, the paper is more - or -less clearly written, there is no classic structure: introduction, material, etc. I would like to see short discussion why this finding is so important, in comparison with the literature. More references are also necessary, ex. 

Maricar N, Parkes MJ, Callaghan MJ, Felson DT, O'Neill TW. Where and how to inject the knee--a systematic review. Semin Arthritis Rheum. 2013 Oct;43(2):195-203. doi: 10.1016/j.semarthrit.2013.04.010. Erratum in: Semin Arthritis Rheum. 2015 Apr;44(5):e18. Erratum in: Semin Arthritis Rheum. 2015 Apr;44(5):e18. PMID: 24157093; PMCID: PMC3820023.

Ertilav E, Sarı S, Ertilav D, Aydın ON. Comparison of radiological and clinical results of knee intra-articular injections with two ultrasonography-guided approach techniques: A randomized controlled study. Arch Rheumatol. 2022 Oct 21;38(2):230-237. doi: 10.46497/ArchRheumatol.2023.9382. PMID: 37680515; PMCID: PMC10481686.

Response:

We acknowledge the reviewer's thoughtful comments. The article has been submitted to the "interesting imaging" session rather than the "review" or "original article" category. Consequently, there is no imperative to adhere to the conventional structural format with subtitles such as "introduction," "material and methods," "results," "discussion," and "conclusion."

For reference, we have provided examples of articles in the "interesting images" category from "Diagnostics" at the following links:

(1) https://www.mdpi.com/2075-4418/14/1/108

(2) https://www.mdpi.com/2075-4418/14/1/68

However, in response to the reviewer's concerns, we have incorporated the following subtitles to enhance the organizational structure of our article:

(1) Introduction

(2) Material and Methods

(3) Anatomic Elaboration of Articularis Genus 

(4) Sonoanatomy of Articularis Genus

(5) Implication of Ultrasound Guided Injection

(6) Conclusion

   Moreover, the two recommended references (Maricar N, Parkes MJ, Callaghan MJ, Felson DT, O'Neill TW. "Where and how to inject the knee--a systematic review." Semin Arthritis Rheum. 2013 Oct;43(2):195-203; Ertilav E, Sarı S, Ertilav D, Aydın ON. "Comparison of radiological and clinical results of knee intra-articular injections with two ultrasonography-guided approach techniques: A randomized controlled study." Arch Rheumatol. 2022 Oct 21;38(2):230-237) have been appropriately cited in the revised manuscript with a brief discussion, as follows: “Additionally, a prior systematic review [8] underscored the superior accuracy of image-guided intra-articular knee injections at specific sites, including the superomedial patellar, medial midpatellar, superolateral patellar, and lateral suprapatellar bursae, compared to blind injections. However, there remains a knowledge gap regarding the long-term effectiveness of ultrasound-guided intra-articular knee injections compared to blind injections. A recent randomized controlled trial [9] demonstrated that ultrasound-guided intrapatellar injection led to an early and positive opaque transition to the knee joint in comparison to the suprapatellar approach, despite the clinical effects appearing similar between the two methods. Given that our report concentrates on a concise exploration of the sonoanatomy of the articularis genus and its relevance to guided injection, we have not conducted a formal study assessing the impact of detailing the sonoanatomy on the precision of intra-articular knee injections. This aspect could be investigated in future cadaveric and clinical studies.”

Round 2

Reviewer 2 Report

Comments and Suggestions for Authors

Thank you for revising the manuscript. My concerns have been addressed. I do not have any further questions. 

Reviewer 3 Report

Comments and Suggestions for Authors

Improved, in my opinion now the paper is easier to reads and understand. It is worth publishing.